Metagenome analysis of viruses associated with Anopheles mosquitoes from Ramu Upazila, Cox’s Bazar District, Bangladesh

Li Tao 1
http://orcid.org/0000-0001-8330-5499 Shafiul Alam Mohammad 2
Yang Yu 1
http://orcid.org/0000-0003-2027-2399 Mohammad Al-Amin Hasan 2 3
Rahman Mezanur 4
Islam Farzana 4
http://orcid.org/0000-0003-2712-7960 Conte Matthew A. 1
Price Dana C. 5
Hang Jun 1 jun.hang.civ@health.mil
1 Viral Diseases Program, Walter Reed Army Institute of Research , Silver Spring, Maryland , United States
2 Infectious Diseases Division, International Centre for Diarrhoeal Disease Research , Bangladesh (icddr,b), Dhaka , Bangladesh
3 School of the Environment, The University of Queensland , Queensland , Australia
4 Department of Zoology, Jagannath University , Dhaka , Bangladesh
5 Department of Entomology, Center for Vector Biology, Rutgers, The State University of New Jersey , New Brunswick, New Jersey , United States
Brygadyrenko Viktor
Electronic publication date: 2025 Mar 31
Publication date: 2025
Volume: 13
Electronic Location ID: e19180
Received 2024 Dec 13; Accepted 2025 Feb 25
Copyright: © 2025 Li et al.
Copyright year: 2025
Copyright holder: Li et al.
License: This is an open access article distributed under the terms of the Creative Commons Attribution License, which permits unrestricted use, distribution, reproduction and adaptation in any medium and for any purpose provided that it is properly attributed. For attribution, the original author(s), title, publication source (PeerJ) and either DOI or URL of the article must be cited.
License URL: https://creativecommons.org/licenses/by/4.0/

Keywords: Mosquitoes, Emerging diseases, Molecular epidemiology, Infectious diseases, Zoonotic diseases, Arboviruses, Viral diseases, Febrile vector-borne illness

Funding: Global Emerging Infections Surveillance and Response System (GEIS) Division of the Armed Forces Health Surveillance Center P0157_23_WR and P0109_24_WR This work was supported by the Global Emerging Infections Surveillance and Response System (GEIS), a Division of the Armed Forces Health Surveillance Center, projects P0157_23_WR and P0109_24_WR. The funders had no role in study design, data collection and analysis, decision to publish, or preparation of the manuscript.

==============================
Bangladesh has a warm climate and landscapes favourable for the proliferation of mosquitoes. Mosquito-borne pathogens including malaria and arthropod-borne viruses (arboviruses) remain a serious threat to the public health requiring constant vector control and disease surveillance. From November 2018 to April 2019, Anopheles mosquitoes were collected in three unions in the Ramu Upazila (sub-district) of Cox’s Bazar District, Bangladesh. The mosquito specimens were combined into pools based on date of collection, household ID, and sex. Metagenome next-generation sequencing was conducted to elucidate diversity of virus sequences in each pool. Homology-based taxonomic classification and phylogenetic analyses identified a broad diversity of putative viruses from 12 known families, with additional unclassified viruses also likely present. Analysis of male mosquitoes showed some of these viruses are likely capable of being vertically transmitted. Moreover, many of the assembled virus sequences share homology and phylogenetic affinity with segments in sequenced Anopheles genomes, and may represent endogenous viral elements derived from a past evolutionary relationship between these putative viruses and their mosquito hosts.

Introduction

Mosquitoes are major vectors of arboviruses, many of which are associated with recurring epizootic events. Due to increased social, economic, and tourism activities coupled with global climate change, both mosquito species and mosquito-borne illnesses are expanding worldwide (Fischer et al., 2013; Ma et al., 2022). The medical and socioeconomic burdens are significant and far greater for low and medium income countries (LMIC) in tropical and subtropical regions (World Health Organization & UNICEF, 2017; Chilakam et al., 2023; Roiz et al., 2024). Mosquitoes species is known to transmit a number of pathogens causing febrile vector-borne illnesses (FVBI), including dengue, Zika virus disease, and Chikungunya by Aedes mosquitoes, malaria and O’nyong-nyong virus by Anopheles, and Japanese encephalitis and West Nile fever by Culex (Yang et al., 2021; Faizah et al., 2020; Hernandez-Valencia et al., 2023). Host susceptibility studies demonstrate that a broad range of mosquito species may be capable of carrying FVBI pathogens and transmitting diseases to human or animals (de Wispelaere, Despres & Choumet, 2017). Recently, the use of metagenomic and metaviromic next-generation sequencing (mNGS) surveys and bioinformatic analyses has yielded a large and increasing number of both known and novel virus sequences associated with mosquitoes and other arthropod insects in recent years (Hernandez-Valencia et al., 2023; Shi et al., 2016; Fauver et al., 2016; Carissimo et al., 2016; Li et al., 2015; Zhang et al., 2022; Xu et al., 2022; Sadeghi et al., 2018).

Bangladesh, a country in South Asia, has large areas of wetland including the largest river delta on Earth, the Ganges Delta. With its warm and humid tropical climate, abundant freshwater territories, and one of the world’s highest population densities, Bangladesh has been a region with a high prevalence of tropical diseases including endemic malaria and dengue (Hossain et al., 2023; Jibon et al., 2024). A better understanding of mosquito species composition, prevalence, breeding environment and infection status (including microbes and viruses) is increasingly essential for prevention and control of known FVBI and preparation for emerging pathogens in Bangladesh and other LMICs.

The work by Alam et al. (2010) showed the presence of over 30 anopheline species in Bangladesh (Al-Amin et al., 2023). The majority of malaria cases in Bangladesh are localized to southwestern districts including Cox’s Bazar and Chittagong Hill Tracts. Studies focusing on the biology and vectorial capacity of Anopheles mosquitoes and surveillance of malaria parasites have been conducted in the region, however a broader survey of the viruses borne in these mosquitoes has not been conducted in this region. In this study, we utilize mNGS analysis of male and female Anopheles mosquitoes from malaria endemic Ramu Upazila (sub-district) of Cox’s Bazar to characterize multiple viral sequences in both sexes. The acquired data implicated the modes of acquisition of mosquito-associated viruses and their potential shared ancestry with endogenous viral elements (EVE) in arthropod insect genomes (Wallau, 2022; Suzuki et al., 2017).

Materials and Methods

Study area

The mosquito collection was carried out in Ramu Sub-district of Cox’s Bazar District, Bangladesh from November 2018 to April 2019. The site is situated between 21°17′ and 21°36′ north latitudes and in between 92°00′ and 92°15′ east longitudes (Fig. 1); it covers approximately 391.91 km2 with a population of 266,640 people at a density of 680.7/km2 according to a 2011 census. Ramu is bounded by Chakaria and Cox’s Bazar sub-districts on the north, Naikhongchhari and Ukhia sub-districts on the south, Naikhongchhari sub-district on the east, Cox’s Bazar Town and the Bay of Bengal on the west. The landscape comprises paddy fields on the plains and unused shrub lands or teak plantations in hilly areas. Entomological investigations were conducted in the following three villages: Horitola of Rashidnagor union (21°29′817″ N, 92°6′082″ E), VIP Pahar and Forest Office Purbopara of Jowarianala union (21°29′028″ N, 92°6′825″ E) and Moheshkum of Kawerkop union (21°26′062″ N, 92°9′294″ E).

Figure 1 Map of sample collection sites in Bangladesh.

Anopheles specimens were collected in Ramu, an upazila of Cox’s Bazar District in the Division of Chittagong, Bangladesh. The dots show GPS coordinates for mosquito traps inside or outside the households.

Mosquito collections

Adult active mosquitoes were collected only from human dwellings using battery operated CDC miniature light traps Model 1012 (John W. Hock Company, Gainesville, FL, USA) between hours 1800 to 0600 following the procedure described by Alam et al. (2010). Traps were set both inside and outside the main living room, i.e., indoor and outdoor, respectively. In each month, 18 traps were operated. Six traps (three indoor and three outdoor) were installed per night in each village. Thus, in a 6-month collection period a total of 108 traps were installed at the study sites.

Resting adult mosquitoes were collected by pyrethrum spray catches (PSCs) between hours 1900 to 1000 (Ndiath et al., 2011). A single bedroom in each house was selected and all windows, doors, and other escape passages of that room were sealed. Pyrethrum insecticides (the aerosols generally used for domestic purposes) were sprayed in the room for 9–12 s (by dividing the room into 3–4 parts based on the size of the room) and the door was closed for approximately 15 min. Immobilized mosquitoes were collected using mouth aspirator with HEPA filter and kept in plastic vials over the course of 15–25 min. Windows and doors were re-opened for proper ventilation and all members of the house were requested to remain outside for 15 min. Each month, five PSCs were carried out per village. In total, 15 PSCs were conducted spanning three villages per month over 6 months, for a total of 90 PSCs.

Collected Anopheles mosquitoes were morphologically identified using standard taxonomic keys (Rattanarithikul et al., 2006) and kept in individual vials containing silica gel desiccant for storage at room temperature at the International Centre for Diarrhoeal Disease Research, Bangladesh (icddr,b). The room temperature stored samples were shipped to Walter Reed Army Institute of Research on July 25, 2019, and stored under −80 °C until extraction for nucleic acids purification.

Nucleic acids extraction and purification from mosquitoes

Male Anopheles mosquitoes or female Anopheles mosquitoes with no visible engorged blood meal (unfed female) were sorted into pools of up to ten individuals of same sex and species from the same collection date and household and transferred into a 2-ml screw-cap microcentrifuge tube with one 3-mm glass bead and two 1-mm glass beads (Sigma-Aldrich, Burlington, MA, USA). For each pool, 0.6 ml of cell growth medium, Eagle’s Minimum Essential Medium (Quality Biological, Inc., Gaithersburg, MD, USA) supplemented with 10% fetal bovine serum (Thermo Fisher Scientific, Waltham, MA, USA) was added, followed by homogenization on a Mini-Beadbeater 16 (BioSpec Products, Inc., Bartlesville, OK, USA) for 1 min and then centrifugated at 4 °C at 8,000 × g for 20 min. Prior to extraction, free nucleic acids were first digested by adding 12 µl of enzyme mixture, containing 10.4 units of DNase I (New England BioLabs, Ipswich, MA, USA), 53.6 units of Benzonase (Sigma-Aldrich, Burlington, MA,USA) and 2.24 units of RNase A (Qiagen, Venlo, USA) into each well of a 96-well KingFisher deep-well plate (Thermo Fisher Scientific, Waltham, MA, USA). For each sample, 188 µL of clear supernatant was transferred into the deep-well plate, incubated at 37 °C for 90 min and then proceeded to nucleic acid extraction on KingFisher Flex magnetic bead purification system using MagMAX Viral/Pathogen (MVP) DNA/NA kit (Thermo Fisher Scientific, Waltham, MA, USA) per manufacturer’s protocol.

Random amplification, metagenome sequencing and data analysis

Unbiased reverse transcription and PCR cDNA amplification was conducted as described previously (Sanborn et al., 2021; Hang et al., 2012). In brief 9.7 µl of purified RNA was pre-treated with DNase I and then annealed with random hexamer primers at 65 °C for 5 min. SuperScript III Reverse Transcriptase (Thermo Fisher Scientific, Waltham, MA, USA) and Platinum Taq DNA Polymerase High Fidelity (Thermo Fisher Scientific, Waltham, MA, USA) were used in a reverse transcription and PCR amplification, respectively. The thermal cycler program for PCR was: 94 °C for 3 min to activate the Taq DNA polymerase followed by 35 cycles of amplification at 94 °C for 30 s, 55 °C for 30 s and 72 °C for 1 min. The random amplicons were purified using AMPure XP beads (Beckman Coulter, Brea, CA, USA) and quantified using Quant-iT PicoGreen dsDNA Assay Kit (Thermo Fisher Scientific, Waltham, MA, USA). A NGS library was prepared using 25 ng of random amplicons and the Illumina DNA Sample Prep kit (Illumina, San Diego, CA, USA) following the manufacturer’s protocol. The indexed libraries were quantitated using PicoGreen dsDNA assay and pooled at equal concentrations. Concentration and size distribution for the final library pool was measured using a Tapestation instrument and Agilent DNA 5,000 assay kit (Agilent Technologies, Santa Clara, California, USA). The libraries were sequenced using Illumina MiSeq NGS system and Miseq Reagent Kit v3 (600 cycles) in a 2 × 300 bp paired-end run.

Illumina MiSeq sequence read data were analyzed using an in-house pathogen discovery pipeline (Kilianski et al., 2015), which consists of quality processing, sequence assembly, nucleotide BLAST analyses of both assembled contigs and all remaining unassembled reads against the NCBI GenBank non-redundant nucleotide (nr/nt) database, and taxonomic identification of organisms in each sample. The Chan Zuckerberg ID Short-Read MNGS workflow (https://czid.org/; https://github.com/chanzuckerberg/idseq-workflows), a cloud-based metagenomics platform (Kalantar et al., 2020), was also used for visualization of mNGS results of selected samples. The sequence contigs with a length of 1 kb or greater were retained for further analysis. MAFFT version v7.475 with the L-INS-I strategy was used to align nucleotide contigs and publicly available background sequences (Katoh & Toh, 2008). IQ-TREE version 1.6.12 (Nguyen et al., 2015) with 1,000 ultrafast boostrap replicates (“--bb 1,000”) was used in phylogenetic analysis. IQ-TREE ModelFinder was used to determine the best-fit model. FigTree (http://tree.bio.ed.ac.uk/software/figtree/) v1.4.4 was used for phylogenetic tree visualization and figure generation.

Endogenous viral element (EVE) classification

To assess the shared ancestry of, and screen for potential EVEs within, the 64-contig dataset with sequence lengths >1 kbp, we first retrieved the aligned nucleotide window of all associated BLASTn hits to any viral target with an alignment length ≥ 200 bp and a bitscore ≥100 from the NCBI nt database. Additionally, we conducted a second BLASTn search using our 64 queries against a database comprising all Anopheles mosquito genomes available at the time of analysis (n = 89; as of 21 November 2023) available in the NCBI whole-genome shotgun (WGS) database and extracted any hits as above. Each set of extracted hits (viral and mosquito) was clustered at 99% nucleotide identity using cd-hit-est (Fu et al., 2012) to remove database sequences that were returned in multiplicity and thus redundant due to closely related query sequences and overlapping BLAST homology windows. These viral and mosquito-derived representative sequences were then aligned together with the 64 viral contig queries using mafft-einsi (Katoh & Toh, 2008) and a maximum-likelihood phylogeny was constructed from the alignment using IQTREE2 under automatic model selection with nodal supports assessed using 2,000 rapid bootstrap approximations (Nguyen et al., 2015).

Results

Metagenomic sequencing of Bangladesh Anopheles virome

In total, 150 pools (644 individual mosquitoes) of Anopheles mosquitoes were sequenced; An. vagus comprised over half of the total specimens (54%), with an additional 14 other Anopheles species (42%) and a small number of unidentified An. spp. (4.1%) available for mNGS analysis (Table S1). An. gambiae and An. stephensi mosquitoes were used in concomitant malaria studies and not analyzed in this study. Males comprised 63 pools (293 individuals) and females, 87 pools (356 individuals) of un-engorged mosquitoes. A total of 123.4 million of MiSeq sequence reads were obtained for metagenomic analysis. Among the de novo assembled contigs with 500 or more mapped sequence reads, 107 contigs were identified as viral via BLASTn analysis against the NCBI nr/nt database. The identified virus sequences were found in 53 of the 150 pools (35.3%). The identified viruses were from 12 established viral families, including Dicistroviridae, Flaviviridae, Hantaviridae, Iridoviridae, Narnaviridae, Partitiviridae, Peribunyaviridae, Phasmaviridae, Rhabdoviridae, Sedoreoviridae, Solemoviridae, Totiviridae, and unclassified Bunyavirales, unclassified Riboviria, or unclassified viruses (Fauver et al., 2016; Piegu et al., 2014; Coffey et al., 2014; Truong Nguyen et al., 2022). These 64 viral scaffolds from 40 Anopheles pools with lengths of 1 kb or greater and manually curated (Table 1) were retained for further analysis.

Table 1 Virus sequences identified in Anopheles mosquitoes from Ramu Upazila, Bangladesh.

Virus	Anopheles Sex	GenBank Accession	Sequence length, bp	GenBank alignment hits	Sequence identity (%)	
Ramu Anopheles Bunyavirus strain JLT2P1148	F	OR367413, OR367415, OR367416	1,248, 1,012, 1,192	Anopheles bunya-like virus, MW520385.1; XiangYun bunya-arena-like_virus 14, OL700209.1	67.7, 67.7, 69.9	
Ramu Anopheles Bunyavirus strain JLT2P1640	F	OR367358, OR367359, OR367361	1,306, 1,412, 1,032	Anopheles bunyavirus 1, LC772143.1; Rhodopi bunya-like virus, MW520386.1	82.8, 71.0, 80.84	
Ramu Anopheles Bunyavirus strain JPSC4P0642	F	OR367391, OR367392	1,291, 1,471	XiangYun bunya-arena-like_virus 14, OL700209.1; Anopheles bunya-like virus, MW520385.1	75.7, 68.0	
Ramu Anopheles Bunyavirus strain RPSC1P0827	F	OR367403	2,377	Wuhan mosquito virus 1, NC_031310.1	65.4	
Ramu Anopheles Chaqlike virus strain JLT2P0298	F	OR367374	1,027	Chaq virus-like 2, KX148555.1	67.6	
Ramu Anopheles Chaqlike virus strain KLT5P0685	F	OR367400	1,352	Chaq virus-like 2, KX148555.1	68.5	
Ramu Anopheles Cripavirus strain JLT1P0118	F	OR367340, OR367342	1,652, 1,235	Anopheles C virus, NC_030115.1	68.2, 73.3	
Ramu Anopheles Orthophasmavirus strain JLT2P1148	F	OR367412, OR367414	1,645, 1,380	Wuhan mosquito virus 1, NC_031307.1; Anopheles triannulatus orthophasmavirus, NC_055395.1	68.5, 65.0	
Ramu Anopheles Orthophasmavirus strain JLT3P1178	F	OR367417	1,023	Wuhan mosquito virus 1, NC_031308.1	66.6	
Ramu Anopheles Orthophasmavirus strain JLT6P0591	F	OR367384	1,162	Wuhan mosquito virus 1, NC_031307.1	71.4	
Ramu Anopheles Orthophasmavirus strain JPSC2P0316	F	OR367377, OR367378	1,239, 1,464	Anopheles triannulatus orthophasmavirus, NC_055395.1; Wuhan mosquito virus 1, NC_031310.1	64.9, 65.0	
Ramu Anopheles Orthophasmavirus strain JPSC2P0596	F	OR367385	1,133	Wuhan mosquito virus 1, NC_031308.1	64.6	
Ramu Anopheles Orthophasmavirus strain JPSC2P0600	F	OR367386	1,324	Anopheles triannulatus orthophasmavirus, NC_055395.1	68.5	
Ramu Anopheles Orthophasmavirus strain JPSC3P0617	M	OR367387	1,609	Anopheles triannulatus orthophasmavirus, NC_055395.1	64.3	
Ramu Anopheles Orthophasmavirus strain JPSC3P0619	M	OR367389	1,145	Anopheles triannulatus orthophasmavirus, NC_055395.1	64.5	
Ramu Anopheles Orthophasmavirus strain KPSC2P0929	M	OR367405, OR367406	1,337, 1,338	Anopheles triannulatus orthophasmavirus, NC_055395.1; Wuhan mosquito virus 1, OL700085.1	64.6, 71.2	
Ramu Anopheles Orthophasmavirus strain KPSC3P0946	F	OR367408	1,588	Anopheles triannulatus orthophasmavirus, NC_055395.1	64.9	
Ramu Anopheles Orthophasmavirus strain KPSC5P0962	F	OR367409	1,611	Anopheles triannulatus orthophasmavirus, NC_055395.1	64.3	
Ramu Anopheles Orthophasmavirus strain KPSC6P1026	F	OR367411	1,686	Wuhan mosquito virus 1, NC_031308.1	69.8	
Ramu Anopheles Orthophasmavirus strain RLT1P0007	F	OR367364, OR367365	1,059, 1,590	Wuhan mosquito virus 1, NC_031308.1	67.3	
Ramu Anopheles Orthophasmavirus strain RPSC1P0827	F	OR367402	1,052	Anopheles triannulatus orthophasmavirus, NC_055395.1	71.8	
Ramu Anopheles Orthophasmavirus strain RPSC1P0837	M	OR367404	1,265	Anopheles triannulatus orthophasmavirus, NC_055395.1	64.6	
Ramu Anopheles Orthophasmavirus strain RPSC1P1552	F	OR367418	1,367	Wuhan mosquito virus 1, NC_031308.1	64.7	
Ramu Anopheles Orthophasmavirus strain RPSC6P1604	M	OR367419	1,941	Wuhan mosquito virus 1, NC_031308.1	66.3	
Ramu Anopheles Orthophasmavirus strain RPSC6P1618	F	OR367420	1,532	Wuhan mosquito virus 1, NC_031308.1	65.2	
Ramu Anopheles Rhabdovirus strain JPSC1P1268	F	OR367346	4,793	Cambodia Anopheles rhabdovirus strain Av13, OR479700	98.1	
Ramu Anopheles Rhabdovirus strain JPSC2P1276	F	OR367348	6,184	Cambodia Anopheles rhabdovirus strain Av13, OR479700	98.4	
Ramu Anopheles Rhabdovirus strain JPSC3P1290	M	OR367349	6,957	Cambodia Anopheles rhabdovirus strain Av13, OR479700	97.9	
Ramu Anopheles Rhabdovirus strain JPSC4P1304	F	OR367351	3,630	Cambodia Anopheles rhabdovirus strain Av13, OR479700	98.2	
Ramu Anopheles Rhabdovirus strain JPSC6P1447	F	OR367354	5,989	Cambodia Anopheles rhabdovirus strain Av13, OR479700	98.3	
Ramu Anopheles Rhabdovirus strain RLT1P0728	F	OR367344	3,608	Cambodia Anopheles rhabdovirus strain Av13, OR479700	97.8	
Ramu Anopheles Rhabdovirus strain RLT4P0783	F	OR367345	5,679	Cambodia Anopheles rhabdovirus strain Av13, OR479700	98.1	
Ramu Anopheles Rhabdovirus strain JPSC2P0316	F	OR367376	4,090	Cambodia Anopheles rhabdovirus strain Av13, OR479700	97.7	
Ramu Anopheles Rhabdovirus strain JPSC3P0617	M	OR367388	1,811	Cambodia Anopheles rhabdovirus strain Av13, OR479700	98.1	
Ramu Anopheles Rhabdovirus strain KLT5P0681	F	OR367398	1,548	Cambodia Anopheles rhabdovirus strain Av13, OR479700	67.0,	
Ramu Anopheles Rhabdovirus strain KPSC6P1015	M	OR367410	2,482	Cambodia Anopheles rhabdovirus strain Av13, OR479700	98.2	
Ramu Anopheles Rhabdovirus strain RLT4P1546	F	OR367356	3,945	Cambodia Anopheles rhabdovirus strain Av13, OR479700	97.9	
Ramu Anopheles Rhabdovirus strain RLT5P1547	F	OR367357	6,246	Cambodia Anopheles rhabdovirus strain Av13, OR479700	98.2	
Ramu Anopheles Ribovirus strain JLT2P1640	F	OR367360	1,339	Hubei mosquito virus 1, NC_033014.1	73.8	
Ramu Anopheles Ribovirus strain JLT6P0311	F	OR367375	3,186	Solemoviridae sp., MT138162.1	66.8	
Ramu Anopheles Seadornavirus strain KPSC3P0946	F	OR367407	2,435	Mangshi virus segment 1, KR349187.1	85.7	
Ramu Anopheles Sobemolike virus strain JLT1P0124	F	OR367372	2,678	Solemoviridae sp., MT138162.1	66.2	
Ramu Anopheles Sobemolike virus strain RLT3P0278	M	OR367373	2,560	Solemoviridae sp., MT138162.1	64.1	
Ramu Anopheles Sobemovirus strain RLT1P0007	F	OR367366	1,954	Riboviria sp., MZ375164.1	63.9	
Ramu Anopheles Totivirus strain JLT1P0118	F	OR367341, OR367343	3,120, 1,187	Australian Anopheles totivirus, NC_035674.1, MF073201.1	77.2, 80.1	
Ramu Anopheles Virgalike virus strain JLT3P1702	F	OR367362	1,524	Hubei virga-like virus 23, OL700071.1	69.8	
Ramu Anopheles Virgalike virus strain JPSC4P0630	M	OR367390	2,949	Hubei virga-like virus 21, NC_033192.1	72.1	
Ramu Anopheles Virgalike virus strain JPSC5P0648	F	OR367394, OR367395, OR367396	1,862, 2,902, 3,455	Hubei virga-like virus 21, OL700070.1, NC_033192.1	71.2, 71.3, 73.6	
Ramu Anopheles Virgalike virus strain JPSC6P1447	F	OR367353, OR367355	3,604, 1,680	Hubei virga-like virus 21, NC_033192.1	71.2, 74.1	
Ramu Anopheles Virgalike virus strain KLT5P0681	F	OR367399	6,553	Hubei virga-like virus 21, OL700070.1	73.9	

Anopheles associated viruses in Ramu, Bangladesh

The viral contigs summed to 147,106 bp, with lengths ranging from 6,957 bp to 1,012 bp, with an average length of 2,263 bp and mean size of 1,590 bp. The BLASTn alignment showed that many sequences exhibited low nucleotide sequence identity (under 75%) with known sequences in aligned regions (Table 1). Annotations derived from BLASTn homology indicate these 68 protein coding sequences (CDS) comprise 6 nucleocapsid, 16 hypothetical protein, 19 glycoprotein, and 27 RNA-dependent RNA polymerase genes. The virus sequences obtained in this study are named after the sample collection location, the Ramu subdistrict, and based on sequence similarities with known virus sequences in GenBank (Table 1). The partial sequence for Ramu Anopheles Seadornavirus strain KPSC3P0946 (family Sedoreoviridae, genus Seadornavirus), 2,435 bp in length encoding RNA-dependent RNA polymerase (RdRp), was most related with segment 1 of Mangshi virus strain DH13M041 (KR349187) (Wang et al., 2015), isolated from Culex tritaeniorhynchus in Yunan Province, China, with nucleotide identity of 85.7%. The sequence shares 66.1% nucleotide identity with Banna virus (NC_004211.1) which can infect human, animal and insects (Attoui et al., 2000). Ramu Anopheles Sobemo-like virus strain JLT1P0124, partial genome sequence of 2,678 bp, showed low nucleotide sequence identity of 65% with viruses from family Solemoviridae, which were found associated with arthropods, plants, and possibly bats (Somera et al., 2021).

Metagenomic analysis identified sequences of Orthophasmavirus (order Bunyavirales, family Phasmaviridae) in 24 (65%) out of the 40 pools and sequences of rhabdovirus (order Mononegavirales, family Rhabdoviridae) in 29 pools (72.5%) (Fig. 2). Specifically, the reference genome sequences for RNA-dependent RNA polymerase (L segment, NC_031307), glycoprotein (M segment, NC_031308), and nucleocapsid (S segments, NC_031310) of Wuhan mosquito orthophasmavirus 1 (genus Orthophasmavirus, species Wuhan mosquito orthophasmavirus 1 or Orthophasmavirus wuhanense) shared homology with viruses identified in this study, with nucleotide identities of approximately 70% for L segment and 65% for M segments. Alignment of M segment nucleotide sequences of Ramu Anopheles Orthophasmaviruses from this study revealed a cluster of 14 closely related strains (Fig. 3), with nucleotide identity over 97% among the strains.

Figure 2 Heatmap of viruses identified in Anopheles mosquito pools using metagenome next-generation sequencing (mNGS).

Mosquito pool numbers, e.g., P0946, are shown on top. The host mosquito sex is indicated, female (green square) and male (purple square). The heat map was set to show samples with at least 50 virus sequence reads per million reads.

Figure 3 Maximum likelihood phylogenetic tree of orthophasmavirus (best-fit model: TIM2+F+R2).

The phylogeny is based on nucleotide sequence alignment of 17 orthophasmavirus sequences from this study, three from Belda et al. (2019) and two related GenBank sequences. Available GenBank accession numbers and mosquito sex, female (F) or male (M), and virus names are showed. Node values represent the percentage of ultrafast bootstrap replicate support.

Thirteen rhabdovirus sequences were identified and designated as Ramu Anopheles Rhabdovirus (Fig. 4). All genome sequences, except for strain KLT5P0681, are nearly identical to recently reported Cambodia Anopheles Rhabdovirus (family Rhabdoviridae) (OR479699–OR479701) (Mohamed Ali et al., 2023), with nucleotide identities of >98% and suggests a high prevalence of related Anopheles rhabdovirus in Southeast Asia. The Ramu Anopheles Rhabdovirus strain KLT5P0681, partial genome sequence of 1,548 nt, has low homology with all other rhabdoviruses from this study and GenBank, with nucleotide identity of 65–70% in aligned regions and bitscores under 200. It shares nucleotide identity of 74.1% and alignment score of 413 with sequence LP1.2_CulexRhabdovCamb, which was identified in Anopheles mosquitoes in Cambodia (Mohamed Ali et al., 2023). Other previously recognized rhabdovirus sequences that were identified in our phylogenetic analysis include Beaumont viruses from Anopheles mosquitoes in Cambodia (Mohamed Ali et al., 2023), Beaumont virus strain 6 from Anopheles mosquitoes in Australia (Coffey et al., 2014), Xanthi rhabdovirus isolate RC1 and Evros rhabdovirus 2 isolate RB2 from Anopheles mosquitoes in Greece, and Culex pseudovishnui rhabdo-like viruses from Japan (Faizah et al., 2020).

Figure 4 Maximum likelihood phylogenetic tree of rhabdovirus.

The phylogeny included 17 orthophasmavirus sequences from this study, four from Belda et al. (2019) and seven related GenBank sequences.

Hubei virga-like viruses (clade Riboviria) are a group of unclassified viruses, which contain protein conserved domain cd23251, an RNA-dependent RNA polymerase (RdRp) in the family Virgaviridae of positive-sense single-stranded RNA [(+)ssRNA] viruses. Sequences that are closely related to Hubei virga-like virus sequences were identified in five Anopheles pools (Fig. 5). Four Ramu Anopheles Virgalike viruses share significant sequence similarity with Hubei virga-like virus 21 strain mosHB236537 (NC_033192, associated with unspecified mosquito species in China) (Shi et al., 2016) and isolate XY8801 (OL700070, Anopheles in China), with amino acid sequence identity of about 80–85%. Sequence for Ramu Anopheles Virgalike virus strain JLT3P1702 is more closely related to Hubei virga-like virus 23 isolate XY43688 (OL700071, Anopheles in China), which is phylogenetically distinct from strain 21. Besides, neither Hubei virga-like viruses 21 and 23 nor Ramu Anopheles Virgalike virus sequence from this study has significant nucleotide identity with any other virga-like viruses identified in Culex mosquitoes or other insects.

Figure 5 Maximum likelihood phylogenetic tree of virga-like virus.

The phylogeny included sequences for five Ramu Anopheles Virga-like viruses from this study and three related GenBank sequences.

Virus sequences identified in male Anopheles mosquitoes

Ten of the 50 virus sequences (20%), which have assembled contigs greater than 1 Kb, were present in male Anopheles (Table 1). These include five orthophasmaviruses, three rhabdoviruses, one sobemolike virus, and one virgalike virus (Table 1), all of which were also seen in female Anopheles in this study and other reports (Figs. 3–5).

Endogenous viral elements

Identification of endogenous viral elements (EVEs) is crucial for future metaviromic work, to avoid misidentification as actively replicating viruses. We identified at least 47 out of the 64 filtered Anopheles viral contigs in clades that shared at least one homologous EVE identified in an Anopheles genome assembly using BLAST searches (Table 2) coupled with phylogenetic analysis (Fig. S1) to confirm monophyly. The largest group of such elements contained 17 of our de novo contigs (12 derived from female and five from male Anopheles mosquitoes), each with homology to the Orthophasmavirus strain Wuhan mosquito virus 1 glycoprotein and integrated as EVEs in the genomes of Anopheles epiroticus, An. cracens, and An. maculatus (Fig. S1 clade A). Remaining homologous elements were identified as derived from: (1) virga-like virus (one contig with top BLAST hits to an element encoded on the genomes of An. atroparvus and An. farauti and six with homology to an RdRP-encoding element currently reported only within An. farauti (Fig. S1 clades B & C, respectively); (2) two nucleocapsid-encoding elements with homology to Anopheles Orthophasmavirus shared broadly with An. maculatus, An. stephensi, An. epiroticus, An. minimus, An. moucheti, An. farauti, An. bellator, An. darlingi, An. ziemanni, An. merus and An. cracens (Fig. S1 clade D); the integration of this EVE may thus pre-date the divergence of major Anopheles species; (3) three contigs encoding nucleocapsid proteins with homology to Anopheles Bunyavirus and present in the genomes of An. koliensis, An. sinensis, An. ziemanni, An. coustani and An. nili (Fig. S1, clade E).; (4) a clade containing twelve RdRP-encoding contigs with homology to Cambodia Anopheles Rhabdovirus with evidence of EVE integration in the genomes of An. rivulorum, An. epiroticus, An. maculatus, and An. funestus (Fig. S1 clade F) that derives from a lineage also containing a separate and distinct Rhabdovirus and associated EVE present in An. merus, An. bwambae, An. quadriannulatus and An. moucheti (Fig. S1 clade G); (5) a single contig encoding an RdRP with homology to Anopheles totivirus and broadly present in the genomes of An. cracens, An. farauti, An. maculatus, An. epiroticus, An. marshallii, An. nili, An. vaneedeni, An. funestus, An. paraensis, An. sinensis, An. atroparvus, and An. punctulatus (Fig. S1 clade H). The number of EVEs derived from viruses related to those described here is evidence for ancient and long-standing evolutionary relationships between these viruses and Anopheles mosquitoes.

Table 2 Homology of Ramu Anopheline virus sequences with putative endogenous viral elements (EVE) in Anopheles genomes.

Query sequence of virus	NCBI Accession	Species	WGS contig ID	% identity	aln length (nt)	e-val	EVE Clade (Fig. S1)	
OR367341_F_Ramu_Anopheles_Totivirus_strain_JLT1P0118	JXWZ01004278.1	An. farauti	N/A	71.9	3,036	0	H	
OR367343_F_Ramu_Anopheles_Totivirus_strain_JLT1P0118	CAJZCY010000007.1	An. atroparvus	PTG000018L	75.2	343	2.56E−58	D	
OR367344_F_Ramu_Anopheles_Rhabdovirus_strain_RLT1P0728	KZ063314.1	An. maculatus	scaffold2300	69.9	3,095	0	F	
OR367345_F_Ramu_Anopheles_Rhabdovirus_strain_RLT4P0783	KZ063314.1	An. maculatus	scaffold2300	70.3	3,939	0	F	
OR367346_F_Ramu_Anopheles_Rhabdovirus_strain_JPSC1P1268	KZ063314.1	An. maculatus	scaffold2300	70.2	3,120	0	F	
OR367348_F_Ramu_Anopheles_Rhabdovirus_strain_JPSC2P1276	KZ063314.1	An. maculatus	scaffold2300	69.4	4,694	0	F	
OR367349_M_Ramu_Anopheles_Rhabdovirus_strain_JPSC3P1290	KZ063314.1	An. maculatus	scaffold2300	69.6	4,694	0	F	
OR367351_F_Ramu_Anopheles_Rhabdovirus_strain_JPSC4P1304	KZ063314.1	An. maculatus	scaffold2300	70.3	3,170	0	F	
OR367353_F_Ramu_Anopheles_Virga-like_virus_strain_JPSC6P1447	KI915042.1	An. farauti	supercont2.3	79.0	2,425	0	C	
OR367354_F_Ramu_Anopheles_Rhabdovirus_strain_JPSC6P1447	KZ063314.1	An. maculatus	scaffold2300	70.2	4,202	0	F	
OR367356_F_Ramu_Anopheles_Rhabdovirus_strain_RLT4P1546	KZ063314.1	An. maculatus	scaffold2300	69.7	3,417	0	F	
OR367357_F_Ramu_Anopheles_Rhabdovirus_strain_RLT5P1547	KZ063314.1	An. maculatus	scaffold2300	70.4	4,202	0	F	
OR367359_F_Ramu_Anopheles_Bunyavirus_strain_JLT2P1640	CALSFZ010000005.1	An. nili	atg000005l	74.7	471	1.39E−81	E	
OR367364_F_Ramu_Anopheles_Orthophasmavirus_strain_RLT1P0007	KZ067125.1	An. cracens	scaffold1007	68.4	465	4.12E−36	A	
OR367375_F_Ramu_Anopheles_Ribovirus_strain_JLT6P0311	KZ067125.1	An. cracens	scaffold1007	67.3	715	1.76E−46	A	
OR367376_F_Ramu_Anopheles_Rhabdovirus_strain_JPSC2P0316	KZ063314.1	An. maculatus	scaffold2300	69.9	3243	0	F	
OR367377_F_Ramu_Anopheles_Orthophasmavirus_strain_JPSC2P0316	KZ067125.1	An. cracens	scaffold1007	68.1	708	7.18E−53	A	
OR367378_F_Ramu_Anopheles_Orthophasmavirus_strain_JPSC2P0316	KZ062071.1	An. maculatus	scaffold398	77.6	1,273	0	D	
OR367385_F_Ramu_Anopheles_Orthophasmavirus_strain_JPSC2P0596	KZ067125.1	An. cracens	scaffold1007	68.0	710	4.41E−55	A	
OR367386_F_Ramu_Anopheles_Orthophasmavirus_strain_JPSC2P0600	KB671765.1	An. epiroticus	supercont1.53	67.1	788	4.25E−56	A	
OR367387_M_Ramu_Anopheles_Orthophasmavirus_strain_JPSC3P0617	KB671765.1	An. epiroticus	supercont1.53	67.6	777	2.87E−59	A	
OR367388_M_Ramu_Anopheles_Rhabdovirus_strain_JPSC3P0617	LULA01026087.1	An. maculatus	contig26087	71.6	1,742	0	F	
OR367389_M_Ramu_Anopheles_Orthophasmavirus_strain_JPSC3P0619	KB671765.1	An. epiroticus	supercont1.53	68.2	777	1.77E-66	A	
OR367390_M_Ramu_Anopheles_Virga-like_virus_strain_JPSC4P0630	KI915042.1	An. farauti	supercont2.3	77.0	2,090	0	C	
OR367392_F_Ramu_Anopheles_Bunyavirus_strain_JPSC4P0642	KE525326.1	An. sinensis	scf7180000696034	70.6	299	2.30E-28	E	
OR367394_F_Ramu_Anopheles_Virga-like_virus_strain_JPSC5P0648	CM027234.1	An. atroparvus	chromosome 2L	71.5	1,648	0	B	
OR367396_F_Ramu_Anopheles_Virga-like_virus_strain_JPSC5P0648	KI915042.1	An. farauti	supercont2.3	77.9	2,249	0	C	
OR367398_F_Ramu_Anopheles_Rhabdovirus_strain_KLT5P0681	OX030915.1	An. moucheti	chromosome 2	66.5	928	2.42E−66	G	
OR367399_F_Ramu_Anopheles_Virga-like_virus_strain_KLT5P0681	KI915042.1	An. farauti	supercont2.3	73.8	2,418	0	C	
OR367402_F_Ramu_Anopheles_Orthophasmavirus_strain_RPSC1P0827	KZ062645.1	An. maculatus	scaffold1091	81.0	284	1.10E−68	A	
OR367403_F_Ramu_Anopheles_Bunyavirus_strain_RPSC1P0827	KB663811.1	An. minimus	supercont1.280	77.9	1,734	0	D	
OR367404_M_Ramu_Anopheles_Orthophasmavirus_strain_RPSC1P0837	KZ067125.1	An. cracens	scaffold1007	68.5	712	1.16E−56	A	
OR367405_M_Ramu_Anopheles_Orthophasmavirus_strain_KPSC2P0929	KB671765.1	An. epiroticus	supercont1.53	67.6	788	1.60E−61	A	
OR367408_F_Ramu_Anopheles_Orthophasmavirus_strain_KPSC3P0946	KB671765.1	An. epiroticus	supercont1.53	67.6	771	8.12E−60	A	
OR367409_F_Ramu_Anopheles_Orthophasmavirus_strain_KPSC5P0962	KB671765.1	An. epiroticus	supercont1.53	67.3	788	4.27E−57	A	
OR367410_M_Ramu_Anopheles_Rhabdovirus_strain_KPSC6P1015	KZ063314.1	An. maculatus	scaffold2300	71.5	2,446	0	F	
OR367411_F_Ramu_Anopheles_Orthophasmavirus_strain_KPSC6P1026	KB671765.1	An. epiroticus	supercont1.53	68.1	777	1.12E−64	A	
OR367413_F_Ramu_Anopheles_Bunyavirus_strain_JLT2P1148	KE525326.1	An. sinensis	scf7180000696034	70.6	299	1.94E−28	E	
OR367414_F_Ramu_Anopheles_Orthophasmavirus_strain_JLT2P1148	KZ067125.1	An. cracens	scaffold1007	68.2	708	1.89E−54	A	
OR367417_F_Ramu_Anopheles_Orthophasmavirus_strain_JLT3P1178	KB671765.1	An. epiroticus	supercont1.53	67.9	788	2.34E−64	A	
OR367418_F_Ramu_Anopheles_Orthophasmavirus_strain_RPSC1P1552	KZ067125.1	An. cracens	scaffold1007	68.8	709	5.71E−61	A	
OR367419_M_Ramu_Anopheles_Orthophasmavirus_strain_RPSC6P1604	KZ062645.1	An. maculatus	scaffold1091	81.8	253	1.92E−62	A	
OR367420_F_Ramu_Anopheles_Orthophasmavirus_strain_RPSC6P1618	KZ067125.1	An. cracens	scaffold1007	67.2	894	1.16E−57	A	

Discussion

The application of unbiased NGS in metagenomic analyses of mosquitoes and other arthropods has revolutionized discovery of insect associated viruses (IAV) (Potter-Birriel et al., 2023; Zhang et al., 2019; Batson et al., 2021). Despite the rapid increase and expanse of mNGS data and associated IAV assemblies, sequence-based identification, and classification of novel viruses, the knowledge of IAV in regions of high FVBI burdens is still limited. With the emergence, recurrence, expansion, and escalation of FVBI threats in low and medium income countries (LMICs), comprehensive information regarding IAVs and their interactions with hosts and other organisms (including co-infecting viruses) is desired (Hernandez-Valencia et al., 2023; Lwande et al., 2024). Many of the Anopheles associated viruses identified herein from Bangladesh are likely to be novel given the lack of strong homology to currently described viruses, however others serve to broaden the known range of previously described IAVs. Viruses associated with Aedes, Anopheles, and Culex mosquitoes in a similar study conducted in Cambodia (Mohamed Ali et al., 2023) for example, include rhabdoviruses isolated from An. vagus with 98% nucleotide identity to those identified here. The study by Mohamed Ali et al. (2023) additionally analyzed An. indefinitus (not included in our study) and identified five viral taxa: Dinovernavirus, Orthophasmavirus, unclassified Flavivirus, Rhabadovirus, and Quaranjavirus while non-anopheline species in the genera Aedes and Culex harbored higher-level viral taxa (including Narnaviridae, Solemoviridae, Totiviridae, and unclassified viruses with similarity with Hubei virga-like viruses) that were also recovered here. Our study together with others in South and Southeast Asia expand knowledge of Anopheles viromes while demonstrating a broad and diverse mosquito virome in the region (Mohamed Ali et al., 2023; Nanfack Minkeu & Vernick, 2018).

Most studies on mosquito viromes investigate female mosquitoes. Male mosquitoes were less studied as they do not feed upon vertebrate hosts and thus are not capable of transmitting associated pathogens. In studies that male and female mosquitoes were analyzed, viruses were found in both sexes (Oguzie et al., 2022; Kubacki et al., 2020). Our results (Table 1) identified multiple viral sequences in male Anopheles mosquitoes from Bangladesh. Considering the short lifespan, commonly only a few days for adult male mosquitoes, it is probably that these viruses are mosquito specific and possibly transmitted vertically through generations. The finding of these viruses in males also suggests that the mosquito-vertebrate transmission cycle is not essential for their maintenance. Numerous reports on detection of viruses in the early stages of mosquito life cycle and in adult male mosquitoes demonstrated the natural occurrence of vertical transmission of both insect specific viruses and arboviruses (Peterson et al., 2024; Ferreira-de-Lima et al., 2020).

Arthropod genomes contain a plethora of elements derived from viruses encountered throughout the course of their evolutionary histories (Palatini et al., 2022). These integrated endogenous viral elements (EVEs) may be lineage-specific, quite numerous, and have been hypothesized to contribute to host antiviral defences via piRNA generation and RNAi-mediated adaptive immunity (Wallau, 2022; Suzuki et al., 2017; Barnes & Price, 2023; Tassetto et al., 2019). Analysis of the viral sequences reported in this study in an evolutionary context with potential cognate EVEs is important for future metagenomic work carried out in a similar fashion using Anopheles species without fully sequenced genomes for confirmation. The biology of most mosquito-associated viruses identified using mNGS is largely unknown; a basic understanding of host and virus interactions will help shed light on the functionality of these viruses, their potential pathogenicity, and relevance with known FVBI pathogens that are co-existing and circulating with these viruses in the same region (Hollingsworth et al., 2023). Many viral sequences assembled in this study were found to exhibit homology with viral elements encoded in Anopheles genomes. The homology of metagenomic contigs with EVEs suggests an ancient evolutionary association between related viruses and Anopheline hosts which may have resulted in endogenization events. Given the potential utility of EVEs as part of the insect immune system, further studies regarding the role extant EVEs play in the host mosquito’s ability to tolerate infection with cognate virus, and even tolerate arboviral infection as a means to becoming a more competent vector, are warranted. It must also be noted that identification and classification of EVEs as such currently relies heavily on homology searches to extant vector and viral sequence data that may not adequately reflect the true diversity of environmental viruses and/or vector genomes (Barnes & Price, 2023; Wang et al., 2022). Of note, Fig. S1 clade A contains three contigs (OR367372, OR367373 and OR367366) with strongest homology to Anopheles sobemo-like virus with no BLAST hits to the aforementioned Orthophasmavirus; these contigs remain classified as viral for this reason and may cluster with orthophasmaviral EVEs in our analysis due to phylogenetic artifacts such as long-branch attraction resulting from neutrally-evolving EVE sequence degeneration, or alternatively conflicting signal from multiple viral ORFs on a single polyprotein-encoding scaffold. As more data are incorporated into public databases, some viral/EVE assignments may change commensurately.

There are over 500 Anopheles species in the world; some are vectors for Plasmodium parasites and filarial nematodes, causing malaria and lymphatic filariasis respectively. The only known arboviral pathogen borne by Anopheles is the O’nyong’nyong virus (family Togaviridae, genus Alphavirus), which causes sporadic febrile illness infections with symptoms similar with dengue diseases (Hernandez-Valencia et al., 2023; Saxton-Shaw et al., 2013). In contrast to malaria and other arboviruses borne by Aedes and Culex mosquitoes, the study of viruses associated with Anopheles species has been largely neglected and a more comprehensive and in-depth analysis is clearly warranted (Hernandez-Valencia et al., 2023). FVBIs share similar symptoms and can therefore be misdiagnosed as common diseases such as dengue. mNGS-based technical approaches in pathogen discovery provides broad genomic surveillance of febrile specimens, mosquitoes, other arthropods, and bloodmeals to not only expedite identification of novel insect associated viruses, but also allow predictive assessment of host susceptibility of the viruses and their potential infectivity to humans or animals (Bolling et al., 2015). The enhanced knowledge of associated viruses will have medical and public health relevance, particularly in the expanding and trans-disciplinary One Health approach to global zoonotic disease. Co-infection with varying viral species (whether arbovirus, IAVs, or otherwise) or with bacterial and parasite pathogens may lead to increased disease severity or complexity. On the other hand, some viruses may be capable of being an antagonist against propagation of FVBI pathogens, with potential to be explored as biological countermeasures, as demonstrated in applying Wolbachia to eliminate dengue (AWED) (Anders et al., 2020; Utarini et al., 2021).

Notably, our mosquito specimens were stored at room temperature which may affect RNA integrity and thus lead to missing viral sequences and fragmented viral assemblies. In future studies, collected specimens are recommended to be immediately processed or stored at low temperatures or in appropriate storage buffer to preserve RNA. Moreover, the quantity of Anopheles specimens used in this study may not accurately reflect their prevalence in the region. Continued metagenomic analyses of FVBI-related subjects should include clinical specimens of interest coupled with systematic collection of all relevant regional mosquito (and tick) species. Prolonged outbreaks of dengue in Bangladesh and the threat of Japanese encephalitis (Hossain et al., 2023) will necessitate the inclusion of Aedes and Culex mosquitoes in similar future work.

Conclusions

Using mNGS analysis, a variety of novel viral sequences were identified in female and male Anopheles mosquitoes in Bangladesh. These viruses are from 12 known families with additional still-unclassified representatives. Analysis of male mosquitoes indicates that some of these viruses are likely mosquito specific and thus vertically transmitted. The homology of virus sequences reported here with those present in sequenced Anopheles genomes warrants further detailed examination with regard to their status as endogenous viral elements (EVEs) derived from similar viruses that in fact represent ancient and persisting evolutionary associations between these viral lineages and their mosquito hosts.

Supplemental Information

Supplemental Information 1 Summary of samples and data from the study.

Supplemental Information 2 Maximum likelihood phylogenetic tree illustrating the shared ancestry of putative viral contigs (green; labels prefixed with “QUERY” + (contig id) + top BLAST hit) and endogenous viral elements (red; labels as above) identified in this study, with NCBI vi.

Nodal support values are the result of 2,000 rapid bootstrap approximations. Major clades containing viral scaffolds described here with evidence of associated endogenous viral elements (EVEs) are labeled A – H. Note that some NCBI viral and Anopheles genome sequences may be represented multiple times in the tree due to non-overlapping (or not completely overlapping) BLASTn hits to the same reference returned by multiple queries during the homology search.

Supplemental Information 3 64 sequences with GenBank accession numbers.

We thank Mr. James S. Hilaire, Ms. Nicole R. Nicholas, Mr. Luis Justinianorodriguez, Mr. Tuan K. Nguyen and Ms. April N. Griggs for their assistance in project management, lab management, sample tracking, storage, and retrieval.

Additional Information and Declarations

Competing Interests

The authors declare that they have no competing interests.

Author Contributions

Tao Li performed the experiments, prepared figures and/or tables, and approved the final draft.

Mohammad Shafiul Alam conceived and designed the experiments, analyzed the data, prepared figures and/or tables, authored or reviewed drafts of the article, and approved the final draft.

Yu Yang performed the experiments, prepared figures and/or tables, and approved the final draft.

Hasan Mohammad Al-Amin performed the experiments, prepared figures and/or tables, and approved the final draft.

Mezanur Rahman performed the experiments, prepared figures and/or tables, and approved the final draft.

Farzana Islam performed the experiments, prepared figures and/or tables, and approved the final draft.

Matthew A. Conte conceived and designed the experiments, analyzed the data, prepared figures and/or tables, authored or reviewed drafts of the article, and approved the final draft.

Dana C. Price conceived and designed the experiments, analyzed the data, prepared figures and/or tables, authored or reviewed drafts of the article, and approved the final draft.

Jun Hang conceived and designed the experiments, analyzed the data, prepared figures and/or tables, authored or reviewed drafts of the article, and approved the final draft.

Data Availability

The following information was supplied regarding data availability:

The sequence read data and metadata are available at NCBI SRA: PRJNA917432.

The 64 assembled and annotated virus sequences are available at GenBank (Table 1).

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
