# Peer review of "Metagenome analysis of viruses associated with Anopheles mosquitoes from Ramu Upazila, Cox’s Bazar District, Bangladesh"

_PeerJ, doi:10.7717/peerj.19180_

## Round 0.1 · original submission · Major Revisions

· Academic Editor

Major Revisions

Dear authors, I kindly ask you to revise the manuscript very carefully according to each of the reviewers' comments. I hope that this will allow the reviewers to approve the publication of your manuscript.

Reviewer 1 ·

Basic reporting

The topic explored by the authors is of significant scientific interest, particularly in its examination of viral diversity in Anopheles mosquitoes. However, the manuscript's title and introduction suggest an emphasis on arboviruses, which is misleading given the findings presented. The study did not identify arboviruses, nor does the methodology enable a direct link between the detected viral sequences and circulating arboviruses in the region. The title and introduction should be revised to reflect the actual scope: the identification and characterization of viral sequences in Anopheles mosquitoes.

Additionally, critical methodological limitations undermine the study's outcomes, notably the use of improperly preserved mosquito samples for RNA analysis.

Experimental design

Methodological Concerns:
- Sample Collection and Preservation
The manuscript describes that mosquitoes were stored at room temperature with silica gel desiccant, which is unsuitable for preserving RNA. This is particularly problematic for RNA virus studies, as RNA degrades rapidly without proper stabilization. The authors should clearly state the duration of storage in silica gel and discuss more deeply the potential effects of these practices on RNA integrity and the reliability of the viral sequences detected.

- Pooling details are insufficient.
The manuscript does not specify whether pools were grouped by species or locality (only sex). According to best practices in mosquito research, authors should clarify: 1) The criteria used for pooling mosquitoes. 2) Whether molecular methods were used for species identification in the pools.

- Viral Sequence Taxonomic Assignment
The results and discussion sections state that "sequences may be novel as they exhibited low nucleotide sequence identity." Claims like these lack the rigor required for this type of study. Taxonomic assignment of viral sequences should follow sequence-based taxonomy guidelines established by the International Committee on Taxonomy of Viruses (ICTV). Specifically: 1) Authors must indicate whether the sequences retrieved allow for taxonomic classification within a viral group or justify why this is not possible. 2) Use terms such as "viral sequences" instead of "viruses" to maintain clarity and scientific accuracy. Maintaining a clear distinction between validated viruses, virus-derived sequences, and putative viruses is crucial for the integrity of the field.
Table 1 and related text should be revised to reflect the viral sequence assignments (name them as viral sequences, not viruses). Additionally, the limitations of sequence similarity-based identification should be discussed explicitly.

-Analytical Tools and Parameters
Details about the bioinformatics tools and pipelines used are sparse. The manuscript should:
-Provide the detailed use for tools such as the Chan Zuckerberg ID platform (If you cannot demonstrate its use as part of the methodology that enabled results, it should not be referenced).
- Specify parameters for BLAST searches, including e-values and filtering thresholds.
- Include parameters for tools such as CD-HIT-EST and MAFFT-EINSI in the methods section.
The phylogenetic analysis lacks key details.
Authors should: 1) Specify whether alignments were nucleotide- or amino acid-based. 2) Indicate which viral genes were used for tree construction. 3) Report the substitution model, alignment method, and number of bootstrap replicates

Validity of the findings

-Pooling and Metadata
The manuscript mentions 87 pools of unfed female mosquitoes. This detail should be explicitly stated in the methodology for clarity and reproducibility. Additionally, a supplementary table summarizing metadata for each pool, including read counts, species composition, and locality, would enhance reproducibility and transparency.

-Viral Sequences Identification
The manuscript states that viral sequences were identified in 53 pools. This phrasing is ambiguous—does it mean sequences were found in 53 pools or associated with them? This should be clear. A figure or table showing virus composition by species, and location would help clarify the results.

-Endogenous Viral Elements (EVEs)
While the authors correctly note that EVE identification and classification currently rely heavily on homology searches, methodologies exist that would allow for a more robust characterization of these elements. Given the study's methodology, the authors should limit their claims to noting that some sequences showed identity with sequences in Anopheles genomes. They could then discuss the implications of these sequences for mosquito biology and metavirome research.
Additionally, the assertion that scaffold length and read depth rule out EVEs is unsupported by evidence.

Additional comments

Figures and Tables
Map (Figure 1)
The map is insufficiently detailed. Improvements should include: 1) A scale and geographic boundaries. 2) A legend indicating the localities sampled.

Phylogenetic Figures (Figures 3, 4, 5)
Legends should specify: 1) The viral sequences used. 2) Alignment type (nucleotide or aminoacid). 3) The substitution model used. Also, Including genome organization or fragments used in phylogeny would provide valuable and necessary context.

Table 1
Rename the title to "Viral Sequences Identified in Anopheles Mosquitoes from Ramu Upazila, Bangladesh" to avoid implying that all sequences are confirmed viruses.

Other suggested edits
- Change the title of the "Abundance and Diversity" section to "Viral Sequences Discovery and Characterization," as the study does not analyze diversity or abundance quantitatively.
- Replace terms like "live, extant viruses" with "actively replicating viruses."

·

Basic reporting

The manuscript is well written overall but there are some aspects that should be improved:

This sentence from the Abstract may be misleading: "Analysis of male mosquitoes showed some of these viruses are likely mosquito specific" - finding viral sequences in male mosquito pools indicates that they are not from vertebrate blood, but it does not mean that vertebrates may not have these viruses as well.

In the Introduction, some details on how many Anopheles species occurring in Bangladesh, and which are more common should be given. Also, an overview on viruses known to be transmitted by Anopheles (and also the lack of knowledge about this) is missing in the introduction - it is referred in the discussion, but should be in the introduction as well, to give context to the study.

Lines 44 - 47 - The sentence may mislead to think that these mosquito species transmit all these pathogens. It should be clear the state of the art about the knowledge about which species transmit which pathogens.

Line 49 - citation missing

Line 63 - "however a broader survey of the viruses borne in these mosquitoes has not been conducted." - do you mean on this region, or overall?

Experimental design

Methods are adequate but some aspects should be improved:

It should be made clear if the method used, besides allowing the retrotranscription of RNA followed by amplification of cDNA, does or does not allow to amplify original DNA material, even with a DNase treatment step included. Is that step efficient? This is crucial to understand if the sequences obtained are viral or may come from segments of Anopheles genomes.

Line 77 - comma missing after "north"

Line 87 - "outside the main living room" means outdoors?

Line 89 - 18 traps per month in 6 months should be 144 traps in total. Please explain why there were fewer in total.

Line 103 - It is not "Mosquito extraction" - Maybe change to Nucleic acid extraction and purification from mosquitoes

Line 104 - How were unfed females identified?

Line 122 - is there a step with RNase H? and is there a step of second strand synthesis?

Line 133 - 300 bp + 300 bp

Line 153 - available at the time of analysis

Line 165 - Add "virome" at the end: Metagenomic sequencing of Bangladesh Anopheles virome

Line 166 - Different mosquito species were put in different pools? This should be detailed in the methods. Did the metagenomic sequencing recover mosquito sequences? Was it possible to identify the mosquito species based on the sequences?

Line 171 - how many de novo assembled contigs were produced in total? And how many reads were viral, from the 124 million total reads?

Line 173 - what does it mean: "The identified viral sequences were associated with identified in ..."

Line 178 - The 107 contigs were in 64 scaffolds? I understand that these 64 are detailed in table 1? But are they contigs or scaffolds? In table 1, the alignment lenght should also be added.

Line 182 - use the same formatting in all numbers (with or without comma separating the thousands) in all text

Line 193 - Some of the nucleotide identities are very low (lower than 70%). The reported homologies were the top one , with highest nucleotide identity? How many others there were, with similar homology values?

Line 236 - what do you mean with "associated with male"? Present in males? More prevalent in males than in females?

Line 348 - The information about the exclusion of A. gambiae and An stephensis form this study is important and should be referred in the methods

Figure 2 - legend has repeated parts. Explain what is the scale of yellow to red colour (number of reads per million reads?). What means the grey colour?

Figure 3 - explain what are the values in the nodes. The resolution of the image should be improved.

Validity of the findings

Results are well discussed and all data is provided.

---

## Round 0.2 · Major Revisions

· Academic Editor

Major Revisions

Dear authors, I kindly ask you to improve the manuscript very carefully in accordance with the serious comments of the reviewers.

Reviewer 1 ·

Basic reporting

Lines 45-50.
Although the discussion mentions that Anopheles species are vectors of the Onyong-nyong virus (there is also evidence suggesting they could be vectors of other arboviruses), this information should also be included in the introduction. Since this study focuses on Anopheles and its metavirome, the introduction would benefit from a concise mention of what is currently known about Anopheles mosquitoes and their associated viruses (see review: https://doi.org/10.3390/tropicalmed8100459. This would be more relevant than mentioning other Culicidae species in the introduction.

Line 186.
The title "Abundance and diversity of Anopheles-associated viruses in Ramu, Bangladesh" should be modified. This study does not conduct abundance or diversity analyses, and the results presented under this subsection do not correspond to such analyses. A more appropriate title that accurately reflects the subsection's content would be "Annotation of Viral Sequences."

The Lines 248-253, 256-259, 269-270, 280-282, contain general background information on Endogenous Viral Elements or comments derived from the results. This content does not belong in the results section and should either be removed or relocated to the discussion section.

The Lines 309-312 , 313-316, 320-326 and 338-340 in the discussion, those statements lack supporting references.

Line 308
Please change "identified multiple viruses" to "identified multiple viral sequences".

Lines 309-311
The presence of these sequences in both sexes could also be due to the expression of EVEs in the mosquito population.

Lines 334-352
This section presents background information rather than discussing the results in the context of existing knowledge. While discussing background information is necessary, it should be done in direct relation to the study's findings. Since no viral sequences associated with arboviruses were detected, the discussion should focus on interpreting this result. Potential explanations for the absence of arboviral sequences and its significance in Anopheles metavirome research could be explored.

Line 354
Please modify to "which may affect RNA integrity and thus lead to missing viral sequences and fragmented viral assemblies."

Lines 362-369
The conclusion should consistently state that the study identified viral sequences rather than viruses. Any mention of viruses should be adjusted accordingly throughout the manuscript.

Experimental design

no comment

Validity of the findings

Information related to species pooling is not yet clear:
The researchers mentioned that the "Collected Anopheles mosquitoes were morphologically identified using standard taxonomic keys." Later in the manuscript, they stated that the mosquitoes were kept individually and then pooled in variable numbers (Table S1). Did the researchers record the species composition of each pool?
The results section states: "A total of 644 individual Anopheles mosquitoes were sequenced; An. vagus comprised over half of the total specimens (54%), with an additional 14 other Anopheles species (42%) and a small number of unidentified Anopheles spp. (4.1%)."
It is important to include, at least as supplementary material, a detailed list of all species that were included in the viral sequence screening. Which species account for the 42% mentioned?

Please clarify.

Reviewer 3 ·

Basic reporting

This is a well reported study, with clear objectives, methods and results outlined in a structured manner. The introduction is a good background on the issues of mosquito-borne diseases, particularly in Bangladesh justifying the need for metagenomic sequencing to study the diversity of viruses associated with Anopheles mosquitoes. Methods enable reproducibility. Results are comprehensively, with new data on virus types identified and possible implications discussed.

Experimental design

Metagenomic NGS has been utilised to investigate the virome of Anopheles mosquitoes in Ramu Upazila. The sample size (644 individual mosquitoes segregated into 150 pools), and sequencing strategy are adequate to provide a broad overview of the virome. Data analysis methods, including sequence assembly, BLAST analyses and phylogenetic analyses are appropriate for the study’s goals and back up the findings.

Validity of the findings

A diverse range of viruses are identified, some of which may be new for Anopheles mosquitoes. Endogenous viral elements are also identified suggesting ancient virus integration events in the mosquito genomes – this is an interesting and significant contribution to mosquito-virus evolution.

Unfortunately, conclusions that the potential for mosquito-specific viruses to be vertically transmitted (and their role in mosquito biology) are tentative. This is because no direct experimental evidence for vertical transmission is presented. Viruses have complex lifecycles that can include both horizontal and vertical transmission pathways. Distinguishing between these in natural populations requires detailed longitudinal or generational studies. The lifecycle stages and interactions of these newly identified viruses within Anopheles mosquitoes are not fully elucidated in the study, leaving the exact nature of the transmission and its biological impacts less defined.

Additional comments

The study mentions that mosquito specimens were stored at room temperature, which may compromise RNA integrity and potentially affect the results. Future studies should consider immediate processing or storage at low temperatures to preserve RNA.
The study focuses on male mosquitoes, and including blood-fed females could provide insights into virus transmission dynamics.
I understand the focus on virus ID and phylogenetic analysis of viruses. It would be interesting to suggest future work on functional studies that might explore the pathogenicity and transmission potential of these viruses in relation to understanding of their role in disease ecology.

---

## Round 0.3 · accepted · Accept

· Academic Editor

Accept

Dear authors, I congratulate you on the acceptance of this article for publication.

Reviewer 1 ·

Basic reporting

The authors presented a descriptive analysis of the metaviromes in Anopheles mosquitoes from Bangladesh. They have satisfactorily addressed the comments raised in previous manuscript reviews. As a result, the title and objectives now accurately reflect the scope and findings of the study. The methodology is appropriate for the defined objective, and the results are presented clearly, effectively conveying the essential information of the study. Additionally, the discussion has an acceptable structure, addressing the challenges and limitations of the study.

Experimental design

There are no additional comments.

Validity of the findings

All comments have been addressed

·

Basic reporting

no comment

Experimental design

no comment

Validity of the findings

no comment

Additional comments

I did not find the responses to my review but I checked that the authors have addressed the points I made.